# What is the right direction for time series anomaly detection benchmarking: evidence from evaluation of linear models

## Abstract

Time series anomaly detection (TSAD) progress has been accompanied by a persistent increase in architectural sophistication. In this work, we revisit this trend and demonstrate that a simple score based on a closed-form solution for an ordinary least squares (OLS) regression model outperforms state-of-the-art deep learning baselines. Through extensive evaluation on both univariate and multivariate TSAD benchmarks, we show that linear regression achieves superior accuracy and robustness while requiring orders of magnitude fewer resources. Our further analysis identifies the types of anomalies that can and cannot be reliably captured by linear models, providing insights into their strengths and limitations. Overall findings indicate that current benchmarkings would benefit from inclusion of simple methods as well as more intricate problems that would do require deep learning-based solutions. Thus, future research should consistently include strong linear baselines and, more importantly, develop new benchmarks with richer temporal structures pinpointing the advantages of deep learning models.

## 1 Introduction

Time series anomaly detection (TSAD) emerges in safety- and reliability-critical applications, including predictive maintenance in industrial IoT, early warning in healthcare monitoring, fraud detection in finance (Zamanzadeh Darban et al., 2024). Data there are complex: often they are high-dimensional, non-stationary and noisy. Motivated by this finding, recent research has shifted toward deep learning methods designed to capture complex temporal patterns.

Early statistical approaches – including autoregressive models (AR) (Rousseeuw & Leroy, 2003), density-based methods such as Sub-LOF (Breunig et al., 2000), and nearest-neighbor search, exemplified by Matrix Profile (Zhu et al., 2018), laid the foundation for anomaly scoring but were soon eclipsed by neural architectures capable of richer feature extraction. A more recent method has introduced deep learning to these core prediction, reconstruction, and density estimation ideas (Zamanzadeh Darban et al., 2024). Prediction-based approaches leverage recurrent and attention mechanisms to forecast future values, e.g., LSTMAD (Malhotra et al., 2015), TimesNet (Wu et al., 2022), and OFA (Zhou et al., 2023). Transformer-based detectors such as the Anomaly Transformer (Xu et al., 2021) and TFAD (Zhang et al., 2022) quantify association discrepancies or exploit hybrid time–frequency features. Reconstruction-based paradigms employ autoencoders (Ng et al., 2011; Malhotra et al., 2016), variational models such as Donut (Xu et al., 2018c) or FCVAE (Wang et al., 2024). Recent additions like TShape and KAN-AD (Zhou et al., 2024) incorporate patch-wise attention and parameter-efficient reasoning, respectively. While increasing benchmark scores, the growing architectural complexity raises several questions. First, incremental gains on standard benchmarks often reflect saturation rather than breakthroughs, especially in the absence of strong baselines. Second, pointwise metrics such as Best-F1 (Xu et al., 2018b; Si et al., 2024) can obscure true event-level quality and reward overfitting. Last but not least, deep detectors are resource-intensive and highly sensitive to hyperparameters, which complicates practical deployment.

While a similar stock of deep learning methods emerged in long-term time series forecasting, researchers there have reported an interesting phenomenon. (Zeng et al., 2023) showed that a one-layer linear model outperforms sophisticated Transformer architectures on long-range forecasting

benchmarks. (Toner & Darlow, 2024) further explore simple models: they tie most of them to an equivalent OLS regression problem, differing only in minor architectural details, and show, that using the available closed-form solution OLS solutions helps to further outperform deep models. However, for anomaly detection, we would expect richer families of possible anomaly types — and therefore sufficiently complex TSAD problems, deep architectures can be justified.

Our findings explore the heavy reliance on architectural sophistication in the TSAD field and its necessity. We revisit the field not by introducing another architecture, but by exploring ordinary least squares (OLS)-based linear regression applied to lagged time-series features for TSAD. The linear model consistently outperforms state-of-the-art deep detectors across both univariate and multivariate TSAD benchmarks, widely used in recent studies (Zhou et al., 2024). Beyond accuracy, OLS-based detection is orders of magnitude more efficient and robust, as it is based on an analytical solution.

Our analysis extends beyond empirical comparisons to clarify why such a simple model can be successful. Drawing on Gaussian process (Williams & Rasmussen, 2006) theory and its connection with the interpolation theory (Zaytsev & Burnaev, 2017), we show that the introduced linear model enjoys the minimal risk under the assumption that the anomaly corresponds to the low conditional density of observations. This finding holds for the wide range of functions, which are dense in the space of continuous functions (Van Der Vaart et al., 2008), and can handle periodicity and noise in sequential observations. Such an approach can be generalized to cases where change points in the function are observed (Saatçi et al., 2010).

In light of these findings, we propose a recalibration of TSAD research: linear baselines must be included in future evaluations, and new benchmarks should feature richer temporal structures that expose the advantages of deep models. Furthermore, we have also revisited the linear model and proposed our small yet effective model, which has achieved SOTA performance on multiple datasets.

Our main claims are the following:

- Our simple linear regression model trained via ordinary least squares (OLS) or reduced-rank regression (RRR) for the past history as features achieves state-of-the-art results in a wide range of univariate and multivariate TSAD benchmarks, consistently outperforming recent deep learning detectors while being orders of magnitude more efficient. Thus, future evaluations in TSAD should include strong linear baselines and develop benchmarks with richer temporal structures to pinpoint the advantages of deep models that originate from the inherent complexity of considered problems.

- The major source of improvement for the introduced model is the use of closed-form analytical solutions for estimating model parameters, which guarantees optimal solutions and eliminates the instability associated with gradient-based optimization.

- Despite the model simplicity, we prove that such models can reliably capture a broad class of anomalies as a conditional density estimator, using a theoretical perspective that links OLS-based autoregression to Gaussian process realizations, the first time according to our knowledge.

## 2 BACKGROUND & LITERATURE REVIEW

TSAD is commonly organized into three families: statistical, prediction-based, and reconstruction-based. Each making distinct assumptions about how normality is modeled and how deviations should be scored. Statistical methods monitor local density or neighborhood structure; prediction-based methods forecast the next value and alarm on large residuals; reconstruction-based methods learn an autoencoding of normal behavior and flag poorly reconstructed windows.

**Statistical.** Sub-LOF (Breunig et al., 2000) flags density deviations locally; SAND (Boniol et al., 2021) clusters subsequences by shape in streaming settings; Matrix Profile (Zhu et al., 2018) scores each window by its nearest-neighbor distance. They are lightweight but can struggle with high-dimensional multivariate drift.

**Prediction-based.** Classical AR (Rousseeuw & Leroy, 2003) models provide robust linear baselines; LSTMAD (Malhotra et al., 2015) captures nonlinear dynamics; TimesNet (Wu et al., 2022)

brings 2D "vision-style" temporal variation; OFA (Zhou et al., 2023) frames many TS tasks under a pretrained LM; Transformer detectors like Anomaly Transformer (Xu et al., 2021) further quantify association discrepancies; TFAD (Zhang et al., 2022) couples time–frequency decomposition with detection. Recent works KAN-AD (Zhou et al., 2024) boost detection accuracy with orders-of-magnitude fewer parameters – both complement linear readouts by clarifying where deviations arise. Long-horizon forecasters are increasingly repurposed for detection by thresholding forecast residuals. Autoformer (Wu et al., 2021) introduces an Auto-Correlation mechanism with progressive decomposition for long-term forecasting, alleviating pointwise attention bottlenecks and improving periodic pattern capture. ModernTCN (Luo & Wang, 2024) revisits temporal convolutions with a modern, pure-CNN block that scales receptive fields and cross-variable coupling, yielding state-of-the-art tradeoffs across forecasting, imputation, classification, and anomaly detection. In contrast, CATCH (Wu et al., 2024) targets TSAD directly: it patchifies the frequency domain and fuses channels via masked attention to capture fine-grained spectral characteristics and channel correlations – key for heterogeneous multivariate anomalies.

**Reconstruction-based.** Autoencoders such as AE (Ng et al., 2011) and EncDecAD (Malhotra et al., 2016) learn normal reconstructions; TranAD (Tuli et al., 2022) adds adversarial training; Donut (Xu et al., 2018c) uses a VAE for seasonal KPIs; and FCVAE (Wang et al., 2024) strengthens this line by decomposing signals into frequency components to model uncertainty. Industrial deployments such as SRCNN (Ren et al., 2019) blend signal transforms with neural scoring and are widely used in practice.

## 3 METHODS

### 3.1 PROBLEM SETUP

Let $\{y_t\}_{t=1}^{T}$ be a univariate ($d = 1$) or multivariate ($d > 1$) time series with $y_t \in \mathbb{R}^d$. We fix an autoregressive order $p \geq 1$, which specifies how many past observations are used as predictors. To capture temporal dependencies, we define lagged feature vectors

$$x_t = \left(1, y_{t-1}^{\top}, \ldots, y_{t-p}^{\top}\right)^{\top} \in \mathbb{R}^{1+dp},$$

and collect all $T - p$ samples into feature and response matrices as:

$$X = \begin{bmatrix} x_{p+1}^{\top} \\ \vdots \\ x_T^{\top} \end{bmatrix} \in \mathbb{R}^{(T-p) \times (1+dp)}, \quad Y = \begin{bmatrix} y_{p+1}^{\top} \\ \vdots \\ y_T^{\top} \end{bmatrix} \in \mathbb{R}^{(T-p) \times d}.$$

### 3.2 LINEAR MODELING WITH OLS AND RRR

We use a linear predictor based on lagged features $x_t$:

$$y_t = W^{\top} x_t + \varepsilon_t, \quad \varepsilon_t \sim \mathcal{N}(0, \sigma^2 I_d), \tag{1}$$

where $W \in \mathbb{R}^{(1+dp) \times d}$ denotes the matrix of regression coefficients. Anomalies are scored using the squared prediction error, a standard practice in time series anomaly detection:

$$s_t = \left\| y_t - W^{\top} x_t \right\|_F^2. \tag{2}$$

**Ordinary Least Squares (OLS).** Under the Gaussian noise assumption, the maximum likelihood estimate corresponds to minimizing the squared Frobenius norm:

$$\mathcal{L}(W) = \|Y - XW\|_F^2. \tag{3}$$

The minimizer of $\mathcal{L}(W)$ has a closed form, known as the ordinary least squares estimator:

$$\hat{W}_{\text{OLS}} = \arg\min_W \|Y - XW\|_F^2 = \left(X^{\top} X\right)^{-1} X^{\top} Y. \tag{4}$$

In practice, however, the matrix $X^{\top} X$ may be ill-conditioned or singular. To address this, we use a small ridge regularization for numerical stability:

$$\hat{W}_{\text{ridge}} = \arg\min_W \|Y - XW\|_F^2 + \lambda \|W\|_F^2 = (X^{\top} X + \lambda I)^{-1} X^{\top} Y. \tag{5}$$

In our experiments, $\lambda$ is chosen to be very small, so the method remains effectively OLS, while ensuring well-conditioned matrix inversion.

**Reduced-Rank Regression (RRR).** For multivariate outputs, different series often share common temporal patterns, suggesting that the coefficient matrix $W$ in equation 1 may be effectively low-rank. To exploit this latent structure and reduce the number of free parameters, we consider reduced-rank regression (Izenman, 1975):

$$\hat{W}_{\text{RRR}} = \arg \min_{\text{rank}(W) \leq r} \|Y - XW\|_F^2.$$

We can decompose the loss around the OLS solution 4:

$$\|Y - XW\|_F^2 = \underbrace{\|Y - X\hat{W}_{\text{OLS}}\|_F^2}_{\text{constant w.r.t. } W} + \|X\hat{W}_{\text{OLS}} - XW\|_F^2.$$

Since the first term does not depend on $W$, minimizing the loss reduces to finding a rank-$r$ approximation of $X\hat{W}_{\text{OLS}}$ in Frobenius norm.

Let $X\hat{W}_{\text{OLS}} = U\Sigma V^\top$ be the singular value decomposition (SVD). By the Eckart–Young theorem (Golub & Van Loan, 2013), the best rank-$r$ approximation is $U_r\Sigma_r V_r^\top$, yielding

$$\hat{W}_{\text{RRR}} = \hat{W}_{\text{OLS}} V_r V_r^\top,$$

where $V_r V_r^\top$ projects onto the $r$-dimensional subspace capturing the main latent factors. For numerical stability, we replace $\hat{W}_{\text{OLS}}$ with the weakly ridge-regularized estimate $\hat{W}_{\text{ridge}}$ from equation 5.

## 3.3 Computational Complexity

Assuming $T \gg dp$, OLS costs $\mathcal{O}(T(dp)^2)$, while RRR adds a full-rank (in worst case) SVD of $\hat{Y} = X\hat{W}_{OLS} \in \mathbb{R}^{(T-p)\times d}$, costing $\mathcal{O}(Td^2)$. Both methods scale linearly with $T$ and polynomially with $dp$, making them simple, efficient, and practical baselines.

## 3.4 Linear Method Justification

A natural question is what kinds of anomalies can linear autoregression detect? To answer this, we connect it with Gaussian process (GP) modeling and density-based anomaly detection.

Assume the target function is a realization of a stationary GP $f(t) \sim \text{GP}(0, k(t, t'))$, $t \in \mathbb{R}$, so the covariance function that doesn't depend on the location of $t$ and $t'$, but only on their difference $t - t'$. We observe this realization at a uniform grid $D = \{(t = i, y_i)\}_{i=1}^T$. For any $i$, the GP posterior conditional on all other points is

$$p(y_i \mid D_{-i}) = \mathcal{N}(y_i \mid m(i), \sigma^2(i)),$$

with mean $m(i) = \mathbf{k}_i^\top K_{-i}^{-1} \mathbf{y}_{-i}$ and variance $\sigma^2(i) = k(i, i) - \mathbf{k}_i^\top K_{-i}^{-1} \mathbf{k}_i$, where $\mathbf{k}_i = \{k(i, j)\}_{j \neq i}$ and $K_{-i} = \{k(j, j')\}_{j, j' \neq i}$. A natural anomaly score is the negative log-likelihood

$$s(y_i) = -\log p(y_i \mid D_{-i}) = \frac{1}{2}\left[\log(2\pi\sigma^2(i)) + \frac{(y_i - m(i))^2}{\sigma^2(i)}\right]. \tag{6}$$

In anomaly detection we cannot condition on the future. Restricting to the last $h$ lags, $D_{i-h:i-1} = \{(x_j = j, y_j)\}_{i-h \leq j < i}$, corresponds to marginalizing out all other observations, yielding a Gaussian:

$$p(y_i \mid D_{i-h:i-1}) = \int p(y_i \mid D_{-i}) \, p(D_{-i} \mid D_{i-h:i-1}) \, dD_{-i} = \mathcal{N}(y_i \mid m_h(i), \sigma_h^2).$$

with $m_h(i) = \mathbf{k}_h^\top K_h^{-1} \mathbf{y}_{i-h:i-1}$, $\sigma_h^2 = k(i, i) - \mathbf{k}_h^\top K_h^{-1} \mathbf{k}_h$, and blocks $\mathbf{k}_h = \{k(i, j)\}_{i-h \leq j < i}$, $K_h = \{k(j, j')\}_{i-h \leq j, j' < i}$. We purposefully used notation for $\mathbf{k}_h, K_h$ as we don't have the dependence on $i$ for the stationary Gaussian process assumption for uniform observations. This implies

that the mean can be written as $m_h(i) = \boldsymbol{\alpha}_h^\top \mathbf{y}_{i-h:i-1}$, a linear function with coefficients $\boldsymbol{\alpha}_h$ that do not depend on the index $i$. Hence, estimating $\boldsymbol{\alpha}_h$ corresponds exactly to fitting a linear regression on the lagged features (Eq. 1). The anomaly score (Eq. 6) reduces, up to an additive constant, to the squared prediction error:

$$s(y_i) \sim (y_i - m_h(i))^2,$$

recovering exactly the linear model-based anomaly score (Eq. 2). A key insight from this derivation is that the GP-based anomaly score, when restricted to a fixed window, is equivalent to the squared error of a linear model, regardless of the underlying kernel's complexity. This means that the rich class of anomalies detectable by a full density GP is, in the finite-history setting, ultimately captured by a simple linear form.

## 4 RESULTS

### 4.1 EXPERIMENTAL SETTINGS

**Datasets:** To ensure comprehensive coverage of anomaly distributions, we have integrated a diverse suite of both univariate and multivariate benchmarks spanning multiple domains.

*Univariate.* We adopt five annotated datasets, each emphasizing different anomaly types and application contexts:

- *AIOPS (AIO, 2018):* Sourced from five leading Internet firms (Sogou, eBay, Baidu, Tencent, Alibaba), this multidimensional collection comprises system logs, resource metrics, and event traces. It challenges models with evolving distributions, and heterogeneous anomalies ranging from hardware faults to security breaches.
- *UCR (Wu & Keogh, 2021b):* A canonical repository of 203 time-series across domains (such as power-grid, medical sensors, industrial IoT), each containing a single expert-verified anomaly interval. UCR measures a model's generalization across distinct domains and anomaly types.
- *TODS (Lai et al., 2021):* A synthetic suite in which anomalies are injected with precise control over seasonality, trend, and noise parameters. Its ground-truth clarity and tunable complexity enable incisive analysis of design components.
- *NAB (Ahmad et al., 2017):* Streaming data from real-world AWS cloud metrics, social media activity, and IoT sensors, augmented with synthetic sequences. NAB reflects operational detection scenarios where real-time processing and hybrid anomaly sources coexist.
- *Yahoo (Laptev et al., 2015):* Yahoo dataset encompasses both real-world time series and synthetically generated datasets. The real data capture intricate holiday effects and infrastructure migrations, while the synthetic subset is designed to rigorously probe the sensitivity of models to controlled interventions.

Each univariate time series is treated independently: we train a separate model instance per sequence and evaluate on its held-out test split. To ensure fairness and comparability, our training and evaluation protocol follows the EASYTSAD benchmark[1].

*Multivariate.* For the multivariate setting, we rely on five widely used benchmarks covering diverse domains and anomaly characteristics:

- *SMD (Su et al., 2019):* A large-scale dataset of server machine logs from an Internet company. It contains 28 groups of multivariate sensor measurements with annotated anomalies caused by hardware and software faults.
- *MSL* and *SMAP* (Hundman et al., 2018): Both datasets originate from NASA telemetry of spacecraft components. They include dozens of channels monitoring spacecraft systems, with anomalies reflecting system failures and sensor malfunctions.
- *SWAT (Mathur & Tippenhauer, 2016):* Multivariate time series collected from a water treatment testbed, designed to simulate cyber-physical attacks and equipment faults. It is widely used to evaluate anomaly detection in industrial control systems.

---

[1] https://adeval.cstcloud.cn/

- *PSM (Abdulaal et al., 2021):* Real-world server metrics from eBay's production environment. It captures performance anomalies related to distributed system operations and large-scale web services.

For multivariate time series, we follow the standard train-test splits commonly used in the literature. Models are trained on the training set and evaluated on the held-out test set to assess their performance.

**Baselines:** We compare OLS against 21 baselines: SubLOF (Breunig et al., 2000), AR (Rousseeuw & Leroy, 2003), POLY (Li et al., 2007), AE (Ng et al., 2011), LSTMAD (Malhotra et al., 2015), PCA (Aggarwal, 2016), EncDecAD (Malhotra et al., 2016), Donut (Xu et al., 2018c), MatrixProfile (Zhu et al., 2018), SRCNN (Ren et al., 2019), SAND (Boniol et al., 2021), AT (Xu et al., 2021), TFAD (Zhang et al., 2022), TranAD (Tuli et al., 2022), TimesNet (Wu et al., 2022), DLinear (Zeng et al., 2023), TSMixer (Chen et al., 2023), OFA (Zhou et al., 2023), FITS (Xu et al., 2023), FCVAE (Wang et al., 2024) and KANAD (Zhou et al., 2024). For multivariate datasets we compare OLS and RRR against six baselines: Autoformer (Wu et al., 2021), TimesNet (Wu et al., 2022), OFA (Zhou et al., 2023), ModernTCN (Luo & Wang, 2024), CATCH (Wu et al., 2024) and KANAD (Zhou et al., 2024). For each baseline, we use recommended hyperparameters from the original papers.

**Metrics.** To evaluate anomaly detection performance, we report three F1-type metrics in our tables: **F1** – the classical point-adjusted Best F1 score (Xu et al., 2018b), which counts a detection as correct if at least one point in a ground-truth anomalous interval is predicted; **B-F-5** – the $k$-delay Best F1 ($k = 5$) on point-adjusted labels, which requires the first detected point to occur within the first 5 time steps of each anomalous interval; **E-F-5** – the event-level $k$-delay F1 ($k = 5$), treating each contiguous anomalous interval as a single event.

These metrics collectively account for conventional point-level accuracy, temporal responsiveness, and event-level detection, aligning evaluation with practical requirements in real-world anomaly detection scenarios. Full definitions and formal rules for these metrics are provided in Appendix A.

**TSB-AD:** To further validate our models on a large-scale, diverse, and carefully curated benchmark, we evaluate on the TSB-AD (Liu & Paparrizos, 2024) datasets, which comprises over 1000 univariate and multivariate time series across multiple domains. TSB-AD includes both point-level and segment-level anomalies, providing a rigorous test of models' event-level detection and temporal responsiveness. For consistency with prior work on TSB-AD, we report results using the VUS-PR metric (Boniol et al., 2025), while on our main univariate and multivariate benchmarks we continue to report F1, B-F-5, and E-F-5. Our experiments confirm that the relative performance trends observed on the main datasets generalize to TSB-AD.

## 4.2 MAIN RESULTS

Table 1: Model F1-based metrics (↑) on six univariate datasets

| Method | AIOPS | | | NAB | | | TODS | | | UCR | | | WSD | | | Yahoo | | | Average Rank | | |
|---|---|---|---|---|---|---|---|---|---|---|---|---|---|---|---|---|---|---|---|---|---|
| | F1 | B-F-5 | E-F-5 | F1 | B-F-5 | E-F-5 | F1 | B-F-5 | E-F-5 | F1 | B-F-5 | E-F-5 | F1 | B-F-5 | E-F-5 | F1 | B-F-5 | E-F-5 | F1 | B-F-5 | E-F-5 |
| SubLOF | 0.7273 | 0.4994 | 0.2416 | 0.9787 | 0.3169 | 0.0062 | 0.7997 | 0.7169 | 0.5285 | 0.8811 | 0.4539 | 0.5285 | 0.8683 | 0.4917 | 0.3580 | 0.5720 | 0.5560 | 0.4660 | 11.50 | 12.83 | 14.50 |
| AR | 0.9106 | 0.8411 | 0.7262 | 0.9985 | 0.5113 | 0.0881 | 0.7302 | 0.6240 | 0.5462 | 0.7190 | 0.2741 | 0.5462 | 0.9766 | 0.6534 | 0.5702 | 0.7425 | 0.7299 | 0.6810 | 8.33 | 8.00 | 7.83 |
| POLY | 0.7256 | 0.6343 | 0.4172 | 0.9976 | 0.4591 | 0.1034 | 0.4344 | 0.3659 | 0.1719 | 0.4216 | 0.1383 | 0.0049 | 0.6280 | 0.4253 | 0.3113 | 0.5223 | 0.5156 | 0.4172 | 16.50 | 17.33 | 16.50 |
| AE | 0.8934 | 0.8096 | 0.6692 | 0.9896 | 0.4533 | 0.0434 | 0.8472 | 0.7088 | 0.5801 | 0.7157 | 0.2007 | 0.5801 | 0.9742 | 0.6684 | 0.5950 | 0.6847 | 0.6753 | 0.6219 | 10.33 | 10.17 | 9.17 |
| LSTMAD | 0.9395 | **0.8791** | 0.7648 | 0.9907 | 0.4894 | 0.0645 | 0.8295 | 0.7402 | 0.6633 | 0.7763 | 0.3583 | 0.6633 | 0.9875 | 0.6690 | 0.6139 | 0.6096 | 0.6044 | 0.5464 | 6.67 | 5.50 | 5.67 |
| PCA | 0.6172 | 0.3460 | 0.3207 | 0.9752 | **0.5517** | **0.1379** | 0.7401 | 0.5998 | 0.4298 | 0.7103 | 0.2819 | 0.0836 | 0.5970 | 0.3155 | 0.2412 | 0.4463 | 0.3969 | 0.3084 | 16.33 | 12.67 | 14.83 |
| EncDecAD | 0.9121 | 0.8328 | 0.7177 | 0.9903 | 0.5432 | 0.0702 | 0.7107 | 0.5504 | 0.4809 | 0.6759 | 0.2059 | 0.4809 | 0.9829 | 0.6620 | 0.6043 | 0.5682 | 0.5601 | 0.4956 | 11.50 | 9.67 | 9.17 |
| Donut | 0.8588 | 0.7897 | 0.6584 | 0.9829 | 0.5004 | 0.1381 | 0.8648 | 0.7349 | 0.5885 | 0.7619 | 0.2224 | 0.5885 | 0.9642 | 0.6441 | 0.5653 | 0.7302 | 0.7283 | 0.6766 | 9.67 | 9.67 | 7.17 |
| MatrixProfile | 0.1915 | 0.0698 | 0.0125 | 0.7873 | 0.3321 | 0.0079 | 0.5284 | 0.4038 | 0.1288 | 0.7992 | 0.2359 | 0.1288 | 0.1233 | 0.0704 | 0.0134 | 0.3079 | 0.2944 | 0.1926 | 18.33 | 19.00 | 20.83 |
| SRCNN | 0.4176 | 0.1583 | 0.0447 | 0.8945 | 0.3340 | 0.0110 | 0.6140 | 0.4221 | 0.1785 | 0.7424 | 0.2349 | 0.1785 | 0.4187 | 0.1999 | 0.0657 | 0.2289 | 0.1996 | 0.1062 | 17.83 | 18.17 | 19.00 |
| SAND | 0.2823 | 0.0893 | 0.0310 | 0.6731 | 0.2561 | 0.0500 | 0.5336 | 0.5136 | 0.2430 | **0.5637** | 0.1822 | 0.2430 | 0.1822 | 0.1323 | 0.0740 | 0.5646 | 0.5601 | 0.4554 | 18.00 | 15.50 | 18.17 |
| AT | 0.5924 | 0.3500 | 0.2184 | 0.9762 | 0.4263 | 0.0284 | 0.4808 | 0.3184 | 0.1401 | 0.6806 | 0.1368 | 0.1400 | 0.3986 | 0.1323 | 0.0639 | 0.2644 | 0.2517 | 0.1793 | 18.83 | 20.00 | 19.17 |
| TFAD | 0.3486 | 0.1390 | 0.0342 | 0.9543 | 0.3029 | 0.0107 | 0.6131 | 0.4595 | 0.2789 | 0.6317 | 0.1938 | 0.2789 | 0.8462 | 0.5203 | 0.4613 | 0.8134 | 0.8013 | 0.7538 | 15.50 | 15.33 | 14.17 |
| TranAD | 0.8029 | 0.6469 | 0.5786 | 0.9961 | 0.4594 | 0.0332 | 0.5305 | 0.3945 | 0.2174 | 0.6184 | 0.1937 | 0.2174 | 0.7698 | 0.4398 | 0.3813 | 0.6111 | 0.6003 | 0.5417 | 14.33 | 15.50 | 15.17 |
| TimesNet | 0.7853 | 0.6969 | 0.5941 | 0.9901 | 0.4347 | 0.0595 | 0.6602 | 0.4731 | 0.3199 | 0.5999 | 0.1789 | 0.3199 | 0.9015 | 0.5782 | 0.5345 | 0.4976 | 0.4902 | 0.4551 | 15.50 | 15.83 | 13.67 |
| DLinear | 0.9313 | 0.8689 | 0.7865 | 0.9991 | 0.5124 | 0.1089 | 0.8169 | 0.7027 | 0.6038 | 0.7859 | 0.3551 | 0.1362 | 0.9843 | 0.6604 | 0.5920 | 0.7849 | 0.7769 | 0.7304 | 4.50 | 5.50 | 7.33 |
| TSMixer | 0.8997 | 0.8315 | 0.7160 | 0.9984 | 0.5459 | 0.1162 | 0.7939 | 0.6709 | 0.6046 | 0.7330 | 0.2736 | 0.2736 | 0.9777 | 0.6465 | 0.5681 | 0.7338 | 0.7232 | 0.6682 | 8.17 | 8.50 | 9.83 |
| OFA | 0.8402 | 0.7643 | 0.6223 | 0.9851 | 0.4761 | 0.0519 | 0.7023 | 0.5716 | 0.4425 | 0.6780 | 0.1642 | 0.4425 | 0.9782 | 0.6580 | 0.5781 | 0.7520 | 0.7327 | 0.6833 | 11.50 | 11.67 | 10.67 |
| FITS | 0.9125 | 0.8236 | 0.6575 | 0.9942 | 0.4428 | 0.0478 | 0.7772 | 0.5969 | 0.5071 | 0.7570 | 0.3215 | 0.5071 | 0.9714 | 0.6471 | 0.5483 | 0.8074 | 0.7976 | 0.7424 | 7.67 | 9.67 | 10.17 |
| FCVAE | 0.9220 | 0.8486 | 0.7420 | 0.9922 | 0.4936 | 0.1184 | 0.8559 | 0.7339 | 0.6221 | 0.8291 | 0.3269 | 0.6221 | 0.9640 | 0.6553 | 0.5967 | 0.7409 | 0.7389 | 0.6983 | 6.83 | 6.67 | 4.50 |
| KANAD | **0.9458** | 0.8790 | 0.7848 | 0.9911 | 0.5075 | 0.0618 | **0.9469** | 0.8356 | 0.8456 | 0.9050 | 0.5217 | 0.8356 | 0.9867 | 0.6607 | 0.5997 | 0.9597 | 0.9553 | 0.9439 | 2.83 | 3.00 | 3.67 |
| OLS | 0.9418 | 0.8716 | **0.7927** | 0.9979 | 0.5016 | 0.1173 | 0.9100 | 0.8322 | 0.8266 | 0.8332 | 0.5020 | 0.8266 | **0.9877** | 0.7284 | 0.6613 | 0.9695 | 0.9648 | 0.9534 | 2.33 | 2.83 | 1.83 |

Table 1 presents a rigorous comparison of OLS against 21 state-of-the-art baselines across five diverse anomaly detection datasets. OLS achieves the lowest average rank across all three metrics and

Table 2: Model F1-based metrics (↑) on five multivariate datasets

| Method | SMD | | | MSL | | | SMAP | | | SWAT | | | PSM | | | Average Rank | | |
|---|---|---|---|---|---|---|---|---|---|---|---|---|---|---|---|---|---|---|
| | F1 | B-F-5 | E-F-5 | F1 | B-F-5 | E-F-5 | F1 | B-F-5 | E-F-5 | F1 | B-F-5 | E-F-5 | F1 | B-F-5 | E-F-5 | F1 | B-F-5 | E-F-5 |
| Autoformer | 0.5449 | 0.1149 | 0.0061 | 0.8549 | 0.3260 | 0.0218 | **0.9516** | **0.3366** | 0.0147 | 0.8520 | 0.2634 | 0.0073 | 0.9037 | 0.5504 | 0.0193 | 6.00 | 6.00 | 5.40 |
| TimesNet | 0.7137 | 0.1630 | 0.0070 | 0.8475 | 0.2893 | 0.0203 | 0.9368 | 0.3068 | 0.0101 | 0.8823 | 0.3840 | 0.0047 | 0.9650 | 0.6571 | 0.0162 | 5.20 | 4.80 | 6.60 |
| OFA | 0.7181 | 0.1498 | 0.0092 | 0.8749 | 0.3854 | 0.0273 | 0.9472 | 0.2947 | 0.0107 | 0.8936 | 0.3887 | 0.0065 | 0.9699 | 0.6473 | 0.0310 | 3.40 | 4.80 | 4.60 |
| ModernTCN | 0.6999 | 0.1840 | 0.0079 | 0.8627 | 0.3274 | 0.0195 | 0.9163 | 0.2920 | 0.0064 | 0.8875 | 0.3814 | 0.0037 | 0.9650 | 0.6761 | 0.0221 | 5.00 | 4.80 | 6.60 |
| CATCH | 0.7520 | 0.3918 | 0.0781 | 0.7403 | 0.3350 | **0.0873** | 0.8054 | 0.3294 | 0.1075 | 0.9138 | 0.7669 | 0.0471 | 0.9232 | **0.8050** | 0.1357 | 5.60 | **2.60** | 2.60 |
| KANAD | 0.6657 | 0.1564 | 0.0069 | 0.8424 | 0.2732 | 0.0162 | 0.9254 | 0.3027 | 0.0142 | 0.9309 | 0.4263 | 0.0048 | 0.9527 | 0.5417 | 0.0133 | 5.40 | 6.00 | 6.80 |
| OLS | 0.8991 | 0.5231 | 0.3194 | 0.9096 | 0.3968 | 0.0581 | 0.7716 | 0.2843 | **0.1094** | 0.9707 | 0.8188 | **0.1551** | 0.9840 | 0.5767 | 0.3537 | 3.10 | 3.50 | 1.80 |
| RRR | **0.8995** | 0.4859 | 0.3226 | 0.9154 | 0.3970 | 0.0665 | 0.7719 | 0.2707 | **0.1094** | 0.9733 | 0.8647 | 0.1359 | 0.9840 | 0.5767 | 0.3537 | 2.30 | 3.50 | 1.60 |

obtains the highest mean E-F-5 (0.6963) and B-F-5 (0.7335) scores, while achieving the second-highest mean F1 (0.9399), slightly below KANAD (0.9559).

This demonstrates the accuracy of OLS in detecting anomalous events. Our method, relying on a single window size hyperparameter, sets a new state-of-the-art in time series anomaly detection, particularly for dynamic systems with complex local patterns. The consistent gains across metrics and datasets further confirm its suitability for operational deployments. An important observation emerges from the comparison between AR and OLS. While AR and DLinear employ a linear model discrepancy score and estimates parameters via a gradient-based optimization, this approach results in lower performance compared to the closed-form OLS solution, highlighting the advantage of the analytical formulation in accurately capturing anomalies.

As shown in Table 2, these trends remain consistent in multivariate TSAD scenarios as well. OLS consistently outperforms most baselines, confirming that analytical linear reconstruction remains highly competitive even in high-dimensional settings. RRR, which enforces a low-rank structure on the reconstruction model, further improves robustness under strong cross-channel correlations. Together, these results demonstrate that simple analytic models, when properly structured, are powerful and reliable tools for practical multivariate anomaly detection.

To further evaluate robustness on modern real-world challenges, we additionally conduct experiments on the recent and more comprehensive TSB-AD benchmark. Notably, while the original benchmark study highlights the strength of classical statistical baselines such as PCA and POLY, and even claims that deep learning methods dominate in point-anomaly and multivariate scenarios, our results provide a counter-example to this narrative. As shown in the CD diagrams in Figure 1, OLS remains highly competitive in univariate settings, achieving first place on point anomalies and strong performance in overall, although ranked lower on sequence anomalies where long-range temporal modeling is crucial. In the more complex multivariate case, RRR takes the top rank, demonstrating that well-structured linear statistical methods can outperform modern deep architectures even on challenging industrial benchmark datasets.

Table 3: F1 scores (↑) on univariate datasets across anomaly types

| Method | Point-global | | | Point-context | | | Pattern-shape | | | Pattern-seasonal | | | Pattern-trend | | |
|---|---|---|---|---|---|---|---|---|---|---|---|---|---|---|---|
| | F1 | B-F-5 | E-F-5 | F1 | B-F-5 | E-F-5 | F1 | B-F-5 | E-F-5 | F1 | B-F-5 | E-F-5 | F1 | B-F-5 | E-F-5 |
| AR | 0.6822 | 0.3863 | 0.3374 | 0.5411 | 0.5283 | 0.4222 | 0.7615 | 0.4686 | 0.1384 | 0.9478 | 0.6525 | 0.3731 | 0.8496 | 0.2416 | 0.1727 |
| LSTMADalpha | 0.7183 | 0.4167 | 0.3675 | 0.5347 | 0.5347 | 0.4311 | 0.6639 | 0.4338 | 0.1422 | 0.9679 | 0.6878 | 0.2926 | 0.9276 | 0.2263 | 0.0931 |
| AE | 0.7562 | 0.4591 | 0.4063 | 0.3719 | 0.3603 | 0.2616 | 0.7319 | 0.1750 | 0.0907 | 0.8022 | 0.4847 | 0.0710 | 0.7061 | 0.0908 | 0.0071 |
| EncDecAD | 0.6236 | 0.3246 | 0.2501 | 0.3810 | 0.2788 | 0.1906 | 0.4624 | 0.1306 | 0.0081 | 0.8133 | 0.3604 | 0.0965 | 0.3892 | 0.0876 | 0.0075 |
| SRCNN | 0.2399 | 0.1759 | 0.0865 | 0.2819 | 0.2763 | 0.1696 | 0.6076 | 0.1852 | 0.0147 | 0.9436 | 0.4058 | 0.0577 | 0.3892 | 0.2754 | 0.1105 |
| AT | 0.1875 | 0.1341 | 0.0815 | 0.2657 | 0.1621 | 0.0988 | 0.5385 | 0.1061 | 0.0083 | 0.8472 | 0.3038 | 0.0400 | 0.6947 | 0.0946 | 0.0058 |
| TranAD | 0.5910 | 0.2893 | 0.1972 | 0.3154 | 0.3053 | 0.2028 | 0.0658 | 0.0420 | 0.0025 | 0.7032 | 0.3000 | 0.0156 | 0.5519 | 0.0212 | 0.0008 |
| Donut | 0.7064 | 0.4060 | 0.3800 | 0.3733 | 0.3733 | 0.2790 | 0.7846 | 0.2777 | 0.1763 | 0.8627 | 0.4520 | 0.0613 | 0.8336 | 0.0578 | 0.0065 |
| FCVAE | 0.7399 | 0.4400 | 0.3650 | 0.3718 | 0.3573 | 0.2613 | 0.4971 | 0.1551 | 0.0189 | 0.7926 | 0.3815 | 0.0815 | 0.7415 | 0.0986 | 0.0109 |
| TimesNet | 0.7273 | 0.4299 | 0.4132 | 0.3946 | 0.3946 | 0.3478 | 0.6395 | 0.0710 | 0.0037 | 0.7598 | 0.2502 | 0.0182 | 0.6782 | 0.0446 | 0.0037 |
| FITS | 0.7716 | 0.4744 | 0.4490 | 0.7275 | 0.6104 | 0.5091 | 0.8220 | 0.5841 | 0.3999 | 0.9668 | 0.5232 | 0.3281 | 0.8080 | 0.2654 | 0.1209 |
| KANAD | 0.9004 | 0.6131 | 0.5926 | 0.9374 | 0.9360 | 0.9094 | **0.9817** | **0.7342** | **0.7713** | 0.9947 | 0.7187 | 0.3649 | 0.9871 | 0.4444 | 0.3063 |
| OLS | **0.9297** | **0.6325** | **0.6086** | **0.9447** | **0.9431** | **0.9178** | 0.9666 | 0.6796 | 0.4790 | **0.9968** | **0.8027** | 0.3826 | 0.8066 | 0.3770 | 0.2533 |

## 4.3 DISCUSSION ON OLS AND DEEP LEARNING METHODS

Table 3 slices evaluation by anomaly types, including point classes (global and context), and three pattern classes (shape, seasonal and trend) shown in Fig.3 . There are two consistent observations emerge. First, linear autoregression (OLS) dominates *point-type* anomalies, achieving the best scores across all three metrics for both point-global and point-context, with sizeable gaps in event-aware scoring. Second, deep models excel on *shape-type* phenomena, where non-linear deforma-

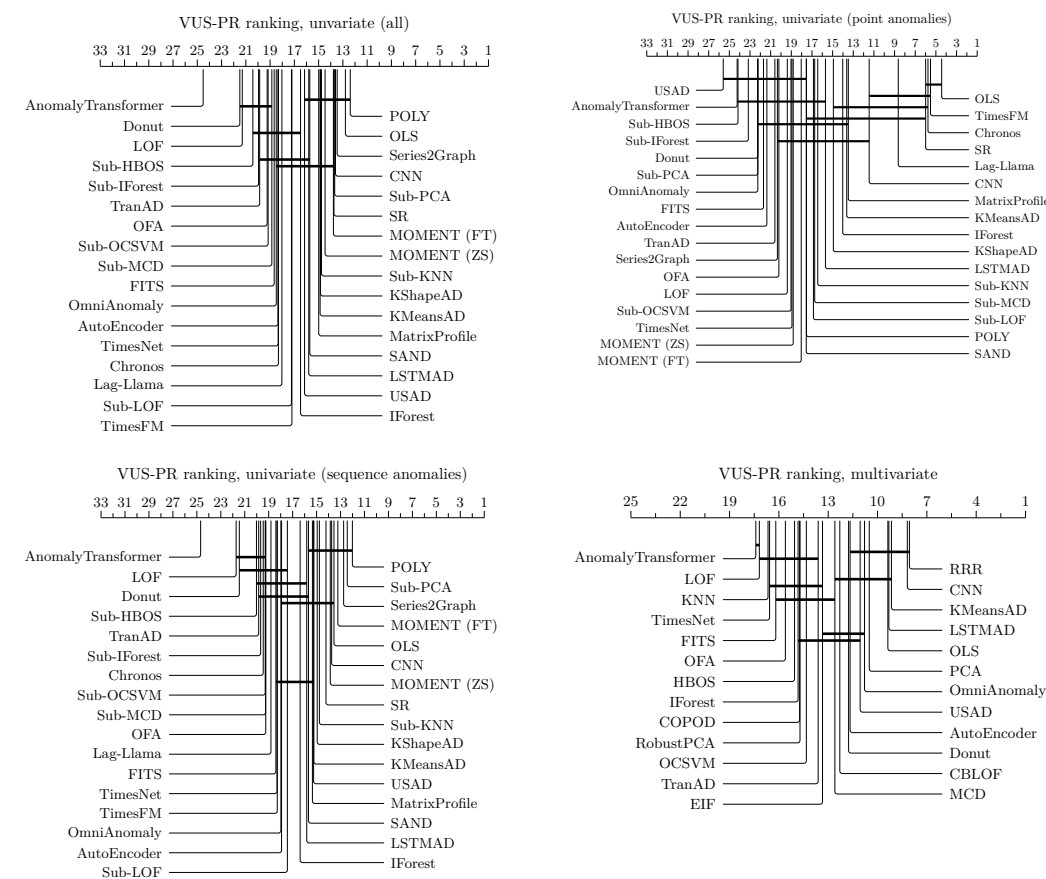

Figure 1: Critical Difference (CD) diagrams comparing the performance of OLS/RRR and baseline methods on the TSB-AD benchmark. Results are shown separately for univariate series (all anomalies), univariate series with point anomalies, univariate series with sequence anomalies, and multivariate series. Methods connected by a horizontal line are not significantly different according to the Nemenyi test ($p < 0.05$).

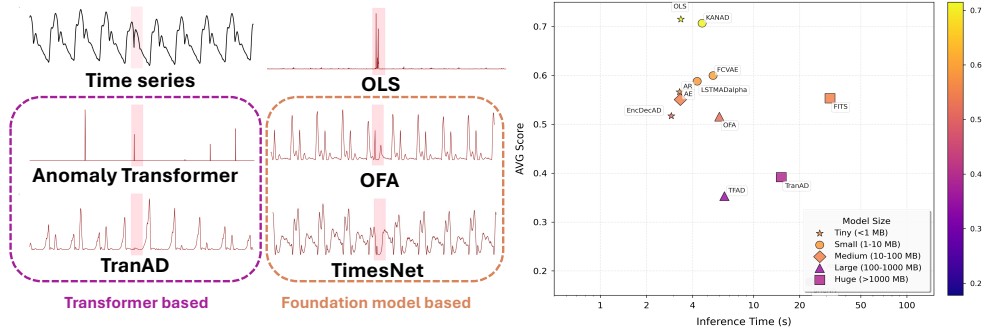

Figure 2: Anomaly detection case study in time series. The original time series is the first black curve, with pink-shaded regions indicating expert-labeled anomaly intervals. Red curves represent the anomaly scores generated by different detection methods using temporal modeling

tions within a contiguous event matter most; here KANAD attains the highest event level detection, while OLS remains competitive on pointwise metrics but lags markedly on E-F-5. Pattern-seasonal is mixed, whereas pattern-trend favors KANAD on all three metrics, suggesting trend-coupled intra-window nonlinearity where parameterized priors over smooth, long-range dynamics help.

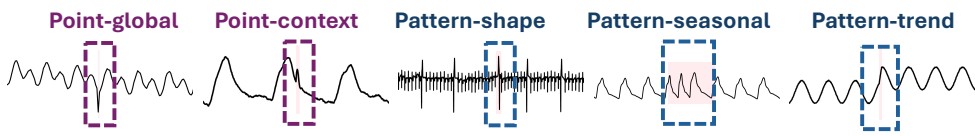

Figure 3: Divide the dataset according to different types of anomalies.

**Why linear models win where they do.** OLS-based lag regression estimates the conditional mean of the next observation from a finite history and scores squared residuals, with a small ridge only for numerical stability. This closed form estimator is the maximum likelihood estimation under Gaussian noise and avoids the optimization instabilities that often plague deep detectors. In our pipeline, this simplicity translates to both robustness and speed. When restricting attention to an $h$-lag window, OLS on lagged features is equivalent to the finite-history posterior mean of a broad family of stationary Gaussian processes; the squared residual is the negative log likelihood under that posterior. Thus, any anomaly that is a low conditional density event under such processes is well captured by a linear predictor with finite memory. Point-global and point-context deviations abrupt spikes, local level shifts, and simple contextual departures fit precisely into this regime, hence the strong linear performance.

**Where deep models buy headroom.** Event level success on pattern–shape and pattern–trend indicates situations where (i) the relevant evidence is distributed across a window or a shpae, (ii) the anomaly is partly invariant to time warps or frequency localized deformations, which looking at only a part of it does not constitute an anomaly, (iii) long range interactions and cross channel couplings fuel non linear effects that exceed finite order linear memory. Architectures that encode patch level nonlinearity, cross channel attention, or frequency aware reasoning can shape a decision surface that better aggregates weak, temporally spread cues into a single event hence KANAD's higher E-F-5 in Pattern-shape and its lead in Pattern-trend .

## 5 CONCLUSION

We revisited time series anomaly detection (TSAD) through the lens of simplicity and showed that ordinary least squares (OLS) regression and reduced-rank regression (RRR) establish a strong new baseline. Across diverse univariate and multivariate benchmarks, OLS consistently surpassed state-of-the-art deep detectors while being vastly more efficient, highlighting that progress in TSAD should be measured against principled baselines rather than architectural novelty.

Our analysis traced these gains to the use of closed-form solutions, which guarantee optimal parameters and avoid the instability of gradient-based methods. Extending to multivariate settings, RRR further improved robustness, with rank and window size tuning reflecting the temporal complexity of each dataset. From a theoretical perspective, we linked OLS-based autoregression to Gaussian process-based conditional density, showing why linear models capture many anomaly types while clarifying where deep models may still be needed.

These findings naturally lead to two imperatives: strong linear baselines must be included in future evaluations, and new benchmarks should feature richer temporal structures that expose when deep architectures truly provide benefits due to their ability to model complex interdependices.

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

## A METRICS

To mitigate the inherent threshold-selection bias in anomaly detection systems (Xu et al., 2018b), we adopt the *Best F1* protocol: given continuous anomaly scores, we sweep a decision threshold over all possible values and report the maximum F1 score obtained. In all tables, the column denoted by **F1** corresponds to this classical *point-adjustment Best F1*: for each ground-truth anomalous interval, the detection is counted as correct as long as at least one time point within the interval is predicted as anomalous, regardless of when the first alarm occurs. This is exactly the "Point Adjustment F1" metric widely used in AIOps benchmarks. However, prior work (Wu & Keogh, 2021a; Xu et al.,

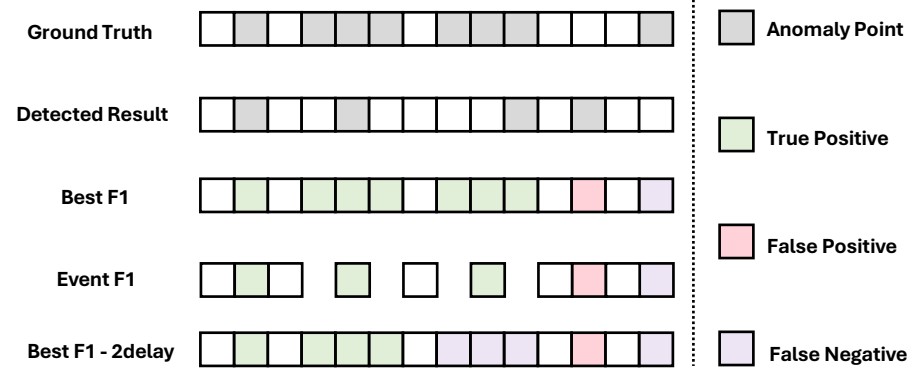

Figure 4: Score Machine.

2018a) has shown that such point-adjusted Best F1 scores are susceptible to artificial inflation, since long anomalous intervals are rewarded multiple times by redundant point-wise detections. More-over, practitioners care about detecting *coherent anomalous events* in a timely manner, rather than labeling every single anomalous point. To better align with this requirement, we additionally employ event-centric and time-sensitive variants. First, we use the *Event F1* score (Si et al., 2024), which treats each maximal contiguous anomalous interval as a single event and measures precision/recall over events; a ground-truth event is considered detected if any prediction overlaps this interval, thereby decoupling event duration from the evaluation of detection capability. Following common practice in time-series anomaly detection for industrial systems, we evaluate models using F1-type scores computed over anomalous *segments* rather than individual time steps.

Second, to explicitly encode the time-critical nature of anomaly detection, we adopt the *F1 k-delay* metric, a stricter evaluation protocol that enforces a delay constraint on successful detections. Under this protocol, an anomalous interval is counted as detected only if the first predicted alarm occurs within the first $k$ time steps after the onset of that interval; alarms raised later than $k$ steps are treated as misses. Concretely, the two delay-aware metrics reported in our tables are denoted by **B-F-5** and **E-F-5**. **B-F-5** is the $k$-delay *Best F1* on point-adjusted labels (also known as $k$-Delay Point Adjustment F1): it uses the same point-adjustment rule as **F1**, but requires the first detected anomalous point to fall within the first $k = 5$ time steps of each ground-truth anomalous interval. **E-F-5** is the event-level analogue, i.e., the $k$-delay *Event F1*: each anomalous interval is treated as a single event, and it is counted as correctly detected only if an alarm is raised within the first $k = 5$ time steps after its onset.

## B RANK SELECTION FOR RRR-BASED APPROACH

Figure 5 shows that the optimal configuration of reduced-rank regression (RRR) is highly dataset-dependent. For example, MSL and SMAP achieve their best F1 scores with relatively low ranks, while SMD and SWAT benefit from higher-rank projections before performance saturates. Similarly, the effect of the temporal window size varies: smaller windows often yield competitive results on datasets with short-range dependencies (e.g., MSL, SMAP), whereas longer histories help capture the broader context required by SMD, PSM, and SWAT. These trends highlight that both the latent rank and the input window must be tuned to the temporal complexity of each dataset rather than treated as universal hyper-parameters.

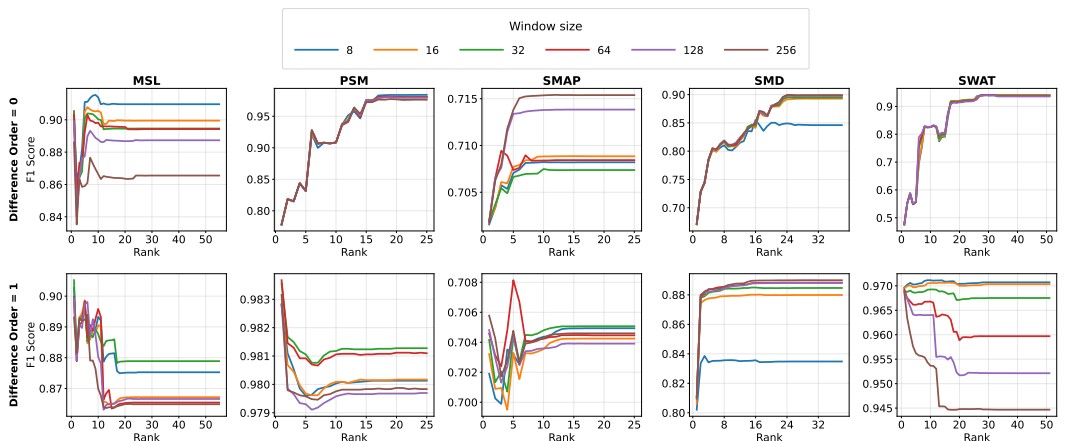

Figure 5: RRR performance across datasets for different window sizes and ranks with Min-Max scaling as a preprocessing step. Full rank (rightmost value) corresponds to OLS baseline
.

Table 4: Optimal hyperparameters for each multivariate dataset

| Dataset | Window size | Rank | Preprocessing | Difference order |
|---------|-------------|------|---------------|------------------|
| MSL | 64 | 16 | Min-Max | 1 |
| PSM | 128 | 32 | Standard | 0 |
| SMAP | 32 | 8 | Min-Max | 1 |
| SMD | 64 | 16 | Standard | 0 |
| SWAT | 256 | 32 | Min-Max | 1 |

# C  HYPERPARAMETER SENSITIVITY

To study the robustness of different methods to the choice of temporal window length, we vary the sliding window size $p \in \{16, 32, 64, 96, 128\}$ while keeping all other hyperparameters fixed. Under the same data splits and evaluation protocol, we report the average anomaly detection scores (using our main metric) for OLS, KANAD, POLY, and FCVAE. As shown in Fig. 4, OLS exhibits strong robustness to the window length. Across $p \in [16, 128]$, the performance of OLS remains in a very narrow band. This indicates that, for the OLS, the exact value of $p$ has limited impact on the detection performance. In practice, a moderately sized window is therefore sufficient to obtain near-optimal results without extensive tuning.

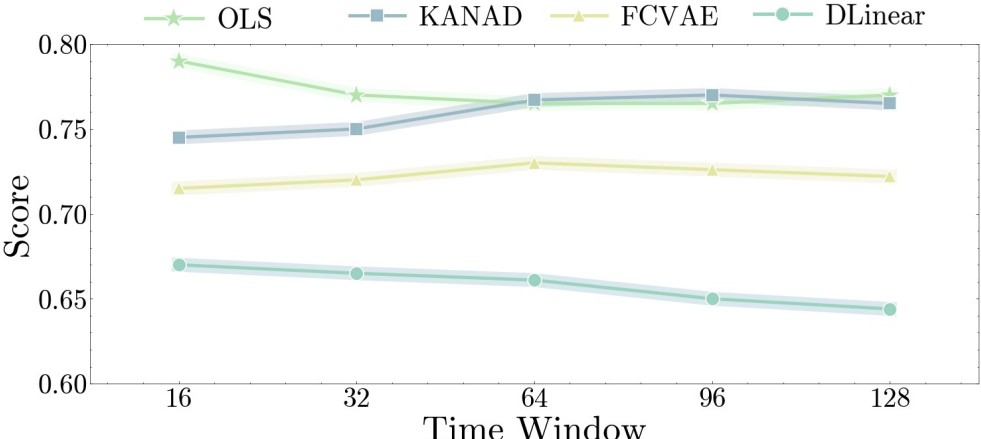

Figure 6: Average anomaly detection scores under different window lengths

