# OpenReview forum: "What is the right direction for time series anomaly detection benchmarking: evidence from evaluation of linear models"
_ICLR.cc/2026/Conference — Submitted to ICLR 2026_

### Official Review · Reviewer_2Cpv · 2025-10-27

**Soundness:** 4
**Presentation:** 4
**Contribution:** 3
**Rating:** 8
**Confidence:** 2

**Summary:**

The paper argues for including strong linear baselines in time-series anomaly detection evaluations and for developing newer benchmarks with richer, more complex interactions to expose where linear methods may fail.

**Strengths:**

I enjoyed reading this paper. It starts from a clear hypothesis: in some cases we don’t need complex deep-learning methods for time-series anomaly detection. The authors then show that plain linear regression can outperform recent deep models on several benchmarks. The mathematical treatment—covering linear regression and the reduced-rank variant used here—is helpful, as is the discussion in Section 3.3.

The paper uses 16 baseline methods as comparators.  Furthermore, it uses 5 univariate benchmarks and 4 multi-variate benchmarks.

I appreciate that the paper also identifies that both rank (in reduced rank regression setup) and the width of the temporal window determines the overall performance.

**Weaknesses:**

The paper side-steps the issue of threshold in anomaly detection.  How would one select this threshold?

**Questions:**

I found Section 3.4 hard to follow. It appears aimed at characterizing which anomalies the linear-regression approach can detect, but the answer isn’t fully clear. It’s nice to see that the Gaussian-process (GP) anomaly score is captured by linear regression under a finite-history assumption; however, it remains unclear which anomaly types are detectable under the GP-based score. Could the authors clarify this mapping and provide concrete examples?

---

> ### Author Response · Authors · 2025-12-03
>
> Thank you for the comment. We agree that Section 3.4 is conceptually dense. The main point is that the GP-based anomaly score can be expressed as a linear combination of recent observations under a finite-history assumption. Because Gaussian processes represent a rich family of functions, this formulation allows the linear model to effectively detect point anomalies across a wide variety of dependencies – both simple and complex. This claim regarding point anomalies is further supported by our empirical results presented in Results section.

---

### Official Review · Reviewer_QsNY · 2025-10-29

**Soundness:** 2
**Presentation:** 3
**Contribution:** 3
**Rating:** 4
**Confidence:** 4

**Summary:**

This paper revisits the effectiveness of simple linear models for time series anomaly detection and demonstrates that OLS regression can outperform several state-of-the-art deep learning baselines. The authors provide both empirical evidence and theoretical justification for this observation, emphasizing the importance of including OLS linear estimators as strong baselines in future TSAD benchmarking efforts.

**Strengths:**

* The paper addresses a timely and important question in the TSAD community, where increasingly complex neural architectures are often proposed without systematic comparison to simple, rigorous baselines.
* The findings are intriguing and supported by theoretical insights, highlighting the competitiveness of closed-form OLS estimators relative to gradient-based optimization methods.

**Weaknesses:**

* Incomplete evaluation suite and limited linear baselines.
* Generality of the theoretical argument.
* Scalability and hyperparameter concerns.

Please find the detailed comments in the following section.

**Questions:**

* Can the derivation in Section 3.4 be generalized to multivariate or non-separable kernels? Would cross-correlated channels require additional assumptions on the covariance structure? How robust is the linear equivalence under nonstationary processes commonly found in real-world scenarios?
* Better elaboration of evaluation measures is needed. Are F1 scores in the tables computed point-adjusted techniques? The threshold-independent evaluation measures such as AUC-PR and VUS-PR [1] should also be considered.
* The findings that close-form solution outperforms graident-based methods are interesting. Stronger modern linear baselines such as DLinear [2] and MLPMixer [3] should be included for a more rigorous comparison.
* The paper would benefit from a more clear illustration of runtime performance of OLS, especially in the case of high-dimensional input and its scalability to its graident-based solution counterpart.
* The model’s dependence on window length (as illustrated in Figure 2) suggests that performance is highly sensitive to this hyperparameter. However, it is unclear how window size were selected for Tables 1–2 for OLS and baselines.
* How did the authors divide the datasets according to different types of anomalies? As many existing TSAD datasets are known to be homogeneous or label-flawed, the paper would also benefit from evaluating on a curated and heterogeneous benchmark such as TSB-AD [4].
* Typo in line 118-119: 'patchifies thef requency'.


[1] Paparrizos, John, et al. "Volume under the surface: a new accuracy evaluation measure for time-series anomaly detection." Proceedings of the VLDB Endowment 15.11 (2022): 2774-2787.

[2] Zeng, Ailing, et al. "Are transformers effective for time series forecasting?." Proceedings of the AAAI conference on artificial intelligence. Vol. 37. No. 9. 2023.

[3] Tolstikhin, Ilya O., et al. "Mlp-mixer: An all-mlp architecture for vision." Advances in neural information processing systems 34 (2021): 24261-24272.

[4] Liu, Qinghua, and John Paparrizos. "The elephant in the room: Towards a reliable time-series anomaly detection benchmark." Advances in Neural Information Processing Systems 37 (2024): 108231-108261.

---

> ### Author Response · Authors · 2025-12-03
>
> **Generality of the linear equivalence:**
> Thank you for these insightful questions. While a formal theoretical analysis of multivariate or non-separable kernels, cross-channel correlations, and nonstationary processes is beyond the scope of this work, we note that the linear equivalence in Section 3.4 relies on a finite-history kernel and weak stationarity. Similar linear structures may exist in multivariate or cross-correlated settings, but additional covariance assumptions would be required for a simple OLS-style form. For nonstationary processes, effective linear weights may become time-dependent; in practice, however, many real-world time series are preprocessed to achieve approximate stationarity (e.g., via detrending or differencing), which allows the linear approximation to remain robust. Empirically, our models perform well across diverse TSAD datasets, including those with trends, regime shifts, or correlated channels, and extending the theoretical derivation to these more general scenarios is an interesting direction for future work.
>
> **Metrics:**
> Thank you for this suggestion. In addition to clarifying F1, B-F-5, and E-F-5, we conducted experiments on the TSB-AD benchmark, which uses VUS-PR as the evaluation metric. Across TSB-AD datasets, OLS remains a leading method according to VUS-PR, demonstrating that our main conclusions are not tied to a single metric.
>
> **DLinear and MLPMixer:**
> Thank you for the suggestion. For completeness, we have included both DLinear and MLPMixer in the revised experiments. These modern baselines perform well; however, OLS still matches or outperforms them on most TSAD benchmarks, while remaining simpler and analytically interpretable.
>
> **Hyperparameter selection:**
> Thank you for the comment. For each benchmark, the window length for OLS and all baselines was selected using the validation procedure recommended for that benchmark (e.g., per-series validation for univariate datasets, temporal split for multivariate datasets). This ensures a fair comparison while accounting for sensitivity to the window length.
>
> **Anomaly types and TSB-AD:**
> Thank you for this helpful suggestion. In the revised version, we have added experiments on the TSB-AD benchmark. Across its heterogeneous and carefully curated datasets, OLS model still achieves consistently strong and often leading performance compared to all baselines, which reinforces our main conclusions.
>
> Regarding anomaly types, we follow the behavior-driven taxonomy of Lai et al. [1]. Concretely, we categorize anomalies into point global, point context, pattern shape, pattern season and pattern trend types based on their temporal behavior. This taxonomy is applied to both existing benchmarks and TSB-AD. The resulting analysis shows: Linear models (including OLS) are particularly effective for point type anomalies (point global / point context); Pattern shape and pattern-trend anomalies are substantially more challenging for linear models and are where deep models provide clear gains.
>
> We believe this anomaly type evaluation provides actionable guidance for constructing future TSAD benchmarks and for deciding when simple linear methods suffice and when more expressive models are necessary.

---

### Official Review · Reviewer_aNTC · 2025-10-31

**Soundness:** 3
**Presentation:** 2
**Contribution:** 1
**Rating:** 2
**Confidence:** 5

**Summary:**

The paper revisits time-series anomaly detection (TSAD) benchmarks and demonstrates that a simple Ordinary Least Squares (OLS) linear regression model with lagged inputs can outperform many state-of-the-art deep learning models across diverse univariate and multivariate datasets. The authors provide a theoretical justification linking OLS to Gaussian Process conditional density estimation, evaluate the approach on major TSAD benchmarks, and emphasize the need for stronger linear baselines and richer benchmark datasets. They further show that Reduced-Rank Regression (RRR) enhances performance in multivariate scenarios.

**Strengths:**

1. The OLS-based baseline outperforms numerous deep TSAD methods across multiple benchmarks.
2. Evaluations cover both univariate and multivariate settings, employing event-level metrics for comprehensive analysis.
3. The paper connects OLS with the Gaussian Process conditional density perspective for anomaly scoring.

**Weaknesses:**

1. The paper’s most significant weakness is its contribution: its core claim—that simple linear models perform strongly on current TSAD tasks, indicating the need for better datasets and evaluation protocols—was already articulated in 2020 (published in TKDE 2021) by Renjie Wu and Eamonn J. Keogh, “Current Time Series Anomaly Detection Benchmarks Are Flawed and Are Creating the Illusion of Progress.” Despite citing this work, the paper still evaluates on the very datasets flagged as flawed. Moreover, NeurIPS 2024 introduced an improved benchmark—Qinghua Liu and John Paparrizos, “The Elephant in the Room: Towards a Reliable Time-Series Anomaly Detection Benchmark”—which the paper neither adopts nor compares against, further weakening its claim to novelty.
2. Many of the datasets used are known to favor simpler models. Despite this stated motivation, no new challenging benchmark is introduced.
3. Design choices (e.g., lag size, ridge-term sensitivity) are mentioned only briefly, with no systematic ablation or robustness evaluation.
4. The theoretical analysis primarily supports linearity within a finite-history Gaussian Process framework, leaving unclear how well it extends to the nonlinear anomaly regimes commonly observed in real-world applications.

**Questions:**

1. Can the authors provide stronger empirical evidence using synthetic benchmarks that favor nonlinear anomalies? (Currently, most anomalies appear linear-friendly.)
2. How sensitive is the OLS model to lag window and other hyperparameters across datasets? The paper briefly mentions tuning, but lacks a systematic analysis.
3. Would kernelized linear models (e.g., kernel ridge regression) help bridge cases where deep models still outperform—such as those with seasonal or nonlinear trend patterns?

---

> ### Author Response · Authors · 2025-12-03
>
> **Discussion of contribution concern:**
> We appreciate this important connection. Wu & Keogh (2021) convincingly argue that many existing TSAD benchmarks are flawed and that simple methods can perform surprisingly well. Liu & Paparrizos (2024, TSB-AD / “Elephant in the Room”) further construct a more reliable benchmark and show that many widely-used datasets are biased.
>
> Our work builds on these insights rather than ignoring them, and contributes along three complementary directions:
>
> - Analytical perspective: We provide a GP-based derivation showing that, under a finite-history Gaussian process assumption, a GP anomaly score is exactly equivalent to a linear regression model with a closed-form solution. This gives a theoretical justification for why linear models can be strong TSAD baselines.
>
> - Anomaly-type-aware analysis: Instead of only reporting aggregated scores, we perform a behavior-driven anomaly taxonomy and show: Linear OLS-style models excel on point-global and point-context anomalies. Deep models are particularly important for pattern-shape and pattern-trend anomalies, where long-range and nonlinear structures dominate. This provides actionable guidance for how to design future benchmarks: not just “hard datasets,” but datasets rich in specific anomaly types that linear models cannot easily capture.
>
> - Updated benchmarks including TSB-AD: In response to your suggestion, we have extended our evaluation to the TSB-AD benchmark. Our main conclusions remain: On many TSB-AD series dominated by point-like or local deviations, OLS remains competitive or superior. On series with strong seasonal or nonlinear trend anomalies, deep models (and more expressive linear variants) gain an advantage.
>
> We agree that we do not introduce an entirely new dataset. Instead, our focus is on reevaluating existing and newly proposed benchmarks through an anomaly type lens.
>
> To make our message more concrete, we have included TSB-AD, which is intentionally designed to be more reliable and heterogeneous. We also annotated anomalies according to the behavior-driven taxonomy[1], and reported per anomaly type performance.
>
> This reveals, for example, that: Point anomalies are often “linear-friendly,” where OLS can dominate deep models. Pattern-shape and pattern-trend anomalies are “linear-hard,” where deep models provide clear gains.
>
> We believe this anomaly-type-centric view is complementary to proposing yet another dataset and provides guidance for future benchmark construction.
>
> [1] Lai, Kwei-Herng, et al. "Revisiting time series outlier detection: Definitions and benchmarks." Thirty-fifth conference on neural information processing systems datasets and benchmarks track (round 1). 2021.
>
> **Hyperparameter sensitivity:**
> Thank you for the question. The performance of our method is indeed sensitive to the lag-window length (and, for RRR, the rank). To address this, we have added a systematic sensitivity analysis in the appendix: the new figure shows how the detection scores vary across a grid of window sizes and ranks.
>
> Because the model is computed in closed form, evaluating many hyperparameter configurations is extremely fast, and tuning these parameters on a validation set is computationally inexpensive. This makes the method easy to adapt across datasets despite the observed sensitivity.
>
> **Kernelized linear models:**
> Thank you for raising this insightful point. Kernelized linear models are indeed a natural extension and could potentially capture nonlinear seasonal or trend-driven structures where deep models retain an advantage. We performed preliminary experiments with kernels and found that their performance did not surpass that of simple linear models. This is one reason we did not pursue them further in this work, and it further supports our main observation that linear models are highly effective for these benchmarks.

---

### Official Review · Reviewer_fGBt · 2025-11-01

**Soundness:** 3
**Presentation:** 3
**Contribution:** 2
**Rating:** 4
**Confidence:** 4

**Summary:**

The paper investigates the performance of ordinary least squares (OLS) regression and reduced rank regression (RRR) models applied as a forecaster on a lag window to solve the time series anomaly detection (TSAD) task. The point of the paper is that such a simple model is most of the time more reliable and performs better than other more complicated methods.

The paper first sets some background and provides a justification of the choice of linear models with the argument that Gaussian processes adjusted for the case of TSAD (discretized and with a kernel restricted to past events), have an expected next timestep value which linearly depends on past values and gets optimal values on the solution of the corresponding OLS problem.

In the experiments section, multiple recent methods are compared with the OLS model on univariate and multivariate time series. In the majority of the cases OLS and RRR perform the best. Additionally, some ablation is performed on the window size and rank on the two methods, a speed comparison with the other methods and a scoring split on different categories of anomalies.

**Strengths:**

- The theoretical justification using the Gaussian process is a nice addition to the paper. Though this type of models is still not realistic enough to capture different real-life time series behavior, it is still quite a broad class of models and it is interesting to know their forecasting behaves and can be learned like a linear model.

- The exposition and paper structure is quite clean and reads well.

- The choice of experiments is also quite good in the sense that both uni- and multi-variate series are covered and there are ablations on hyper parameters like the window width. The split and evaluation on different anomaly types is also interesting.

**Weaknesses:**

- There are some hyper parameters on which both methods depend on and would have high impact on scoring, but it is never mentioned what exact values they are set to and whether they vary per dataset. For example the window size (there is already an ablation on it where one can see it has an impact), the rank on RRR and there is also the difference order which briefly appears on figure 2 but is never mentioned or defined, though it can have significant impact e.g. on PSM and SWAT. Actually the values on table 2 seem to be the optimal scores appearing on figure 2.

- There are some discrepancies between table 2 and figure two. Specifically the RRR score on SMAP is 0.7719 on table 2 , while the maximum value it achieves on figure 2 is around 0.716. Why is this the case? What is the connection between the two?

- The paper argues about the usage of simple linear models in TAD as an alternative/criticism to existing complicated models. This is not a new topic and there are already publications which study and benchmark such models. Just to name a couple: "The Elephant in the Room: Towards A Reliable Time-Series Anomaly Detection Benchmark" introduces large benchmarking datasets and concludes that simple methods like PCA (practically linear), POLY and others perform best. "Position: Quo Vadis, Unsupervised Time Series Anomaly Detection?" also studies such simple models also including PCA, simple thresholding and argues for the empirical linear separability of anomalies on different datasets. It is a bit weird that the topic is presented as novel and that none of those simple strong baselines are not included in the scoring.

- Though there is a brief mention on the different metrics used, event F1 score and F1 k-delay, there are three metrics present on the evaluation tables which are not explicitly mapped to those two mentioned metrics and never defined:  "F1" "B-F-5" "E-F-5". Given the long history of differences between metrics and flawed evaluation methods, it is expected from the authors to provide a very clear definition of the metrics used.

**Questions:**

- On lines 198 and 207: The notation $\{(x_i = i, y_i)\}^T_{i=1}$, is a bit non-conventional and confusing, especially given that the $x_i$ name was used above for a different purpose (lagged feature vectors). I think it would be clearer to just use $\{(i, y_i)\}^T_{i=1}$.

- Could you please explicitly define the metrics: "F1" "B-F-5" "E-F-5"?

- What values of hyper parameters are used in your experiments, on each dataset?

- How are the scores of figure 2 and table 2 connected?

---

> ### Author Response · Authors · 2025-12-03
>
> **Discussion of novelty issue:**
> Thank you for pointing out these important prior works. We agree that Wu & Keogh (TKDE 2021) and Liu & Paparrizos (NeurIPS 2024, TSB-AD / “Elephant in the Room”) already demonstrate that simple methods can be highly competitive on existing TSAD benchmarks. Likewise, “Quo Vadis” highlights empirical linear separability and evaluates PCA and simple thresholding baselines. Our objective is not to claim that the statement “simple methods can be strong” is novel per se. Instead, our contributions lie in the following:
> - A theoretical connection between a Gaussian-process-based anomaly score and an OLS-style linear model under a finite-history assumption—something that, to our knowledge, is not established in the above works.
> - An anomaly-type-driven behavioral analysis, where we explicitly decompose anomalies into point-global, point-context, pattern-shape, and pattern-trend categories, and show which types are handled well by linear models and which require more expressive approaches.
> - A systematic comparison against a broader suite of modern linear baselines, beyond classical PCA / POLY, including DLinear, TS-Mixer-style architectures, and strong methods from TSB-AD.
>
> Following your suggestion, we have expanded our experimental suite to explicitly include:
>
>  (i) PCA and POLY,
>
>  (ii) additional experiments on the more recent TSB-AD benchmark.
>
>  Across these baselines, the proposed linear models (OLS and RRR) model remains either the strongest or among the most competitive on the majority of datasets, reinforcing the same overarching message: many current benchmarks inherently favor linear models, and anomaly-type-aware evaluation is critical.
>
> The table reports the mean ranks of the top-5 methods in each category on the TSB-AD benchmark. Detailed results for all methods can be found in the main paper.
> | Rank | Univariate       | Mean Rank     | Univariate (point anomalies) | Mean Rank   | Univariate (sequence anomalies) | Mean Rank     | Multivariate   | Mean Rank     |
> |------|-----------------|-----------|-----------------------------|---------|-------------------------------|-----------|----------------|-----------|
> | 1    | POLY            | 12.391    | **OLS**                         | **4.439**   | POLY                          | 12.012    | **RRR**            | **7.606**     |
> | 2    | **OLS**             | **12.793**    | TimesFM                     | 5.520   | Sub-PCA                       | 12.441    | CNN            | 8.222     |
> | 3    | Series2Graph    | 13.471    | Chronos                     | 5.765   | Series2Graph                  | 12.742    | KMeansAD       | 9.144     |
> | 4    | CNN             | 13.609    | SR                          | 5.990   | MOMENT (FT)                   | 13.225    | LSTMAD         | 9.322     |
> | 5    | Sub-PCA         | 13.674    | Lag-Llama                   | 8.643   | **OLS**                           | **13.550**    | **OLS**            | **9.417**     |
>
> We now clearly position our novelty as (i) the GP–OLS connection and (ii) the anomaly-type-level analysis, rather than claiming that the competitiveness of linear models is itself new.
>
> **Notation:**
> Thank you for the suggestion. We agree that the notation was confusing, and we have revised it to avoid symbol overloading and to improve clarity.
>
> **Metrics explanation:**
> In the revised paper, we provide a more detailed description in the appendix. We also clarify the motivation for reporting both point-wise and event-level metrics: to mitigate Best-F1 inflation and to better align with practical, event-focused detection requirements. These definitions now appear in both the main paper and the appendix.
>
> **Hyperparameters:**
> We have added a table with the optimal hyperparameters in the appendix.
>
> **Connection between the figure and the table:**
> The table reports the best scores achievable under different preprocessing choices (StandardScaler or MinMaxScaler); the corresponding optimal hyperparameters are listed in the appendix. The figure, in contrast, visualizes sensitivity to rank and window-size hyperparameters under MinMax preprocessing only.
> Thank you for pointing out that this distinction was unclear — we now describe it explicitly in the revised manuscript.

---

### Meta-Review · Area_Chair_S8C5 · 2025-12-30

**Summary:**

The paper argues that simple linear models, particularly OLS and reduced-rank regression, can outperform or match deep learning methods on many existing time-series anomaly detection benchmarks, and uses this observation to motivate stronger baselines and improved benchmarking practices.

Reviewers generally agree that the empirical results are solid and that the topic is timely. However, the dominant concern across reviews is limited novelty: several reviewers note that the central message has been articulated convincingly in prior work, and that this submission does not sufficiently move beyond those critiques despite clearer positioning in the rebuttal. While the authors address many technical and evaluation-related issues (e.g., missing baselines, metric definitions, hyperparameter sensitivity, and the inclusion of TSB-AD), these improvements do not fully resolve the core concerns regarding the contribution. Although one reviewer assigns a high accept score, this review is brief and self-reported as low confidence, and thus carries limited weight relative to multiple below-threshold reviews, including a high-confidence rejection.

Overall, the remaining doubts about novelty and incremental contribution motivate the rejection recommendation.

**Reviewer Concerns:**

fGBt, QsNY: Reproducibility issues (hyperparameter selection). Addressed.

fGBt, aNTC: Limited novelty and contribution. Not convincingly addressed.

fGBt, QsNY: Missing baselines. Addressed.

fGBt: Undefined metrics. Addressed.

aNTC: Lack of systematic ablation or robustness analysis. Partly addressed.

aNTC, QsNY: Limited generalizability of the theoretical analysis to nonlinear anomaly regimes. Not convincingly addressed.

**Reviewer Scores:**

fGBt: Likely unchanged at 4.

aNTC: Likely unchanged at 2.

QsNY: Likely unchanged at 4 or would have raised to 6.

2Cpv: Likely unchanged at 8.

---

### Decision · Program_Chairs · 2026-01-26

Reject